# Review of *Mimusops zeyheri* Sond. (Milkwood): Distribution, Utilisation, Ecology and Population Genetics

**DOI:** 10.3390/plants13202943

**Published:** 2024-10-21

**Authors:** Christeldah Mkhonto, Salmina Ngoakoana Mokgehle, Wilfred Otang Mbeng, Luambo Jeffrey Ramarumo, Peter Tshepiso Ndlhovu

**Affiliations:** 1School of Biology and Environmental Sciences, Faculty of Agriculture and Natural Sciences, University of Mpumalanga, Private Bag X11283, Mbombela 1200, South Africa; 2School of Agricultural Sciences, Faculty of Agriculture and Natural Sciences, University of Mpumalanga, Private Bag X11283, Mbombela 1200, South Africa

**Keywords:** biotechnological applications, ecological roles, ethnomedicinal, genetic diversity, indigenous knowledge, nutrients

## Abstract

*Mimusops zeyheri* Sond. (Milkwood) is an indigenous fruit tree species with considerable ecological, cultural, and nutritional significance that remains underexploited. This review synthesizes current knowledge on its distribution, taxonomy, phytochemistry, ethnomedicinal applications, ecological functions, genetic diversity, and biotechnological potential. A systematic literature search, spanning 1949 to April 2024, yielded 87 relevant publications from an initial 155. *Mimusops zeyheri* plays a crucial role in supporting the cultural traditions and economic activities of Indigenous Southern African Communities. Its distribution encompasses South, East, and Southern Tropical Africa, with substantial populations across South African provinces. Ethnomedicinally, various plant parts treat conditions including wounds, gastrointestinal issues, and diabetes. The leaves (34%) and roots (32%) are used, with infusion (33%) and decoction (31%) as primary preparation methods. Oral administration (70%) is the most common, primarily addressing skin conditions (18%). Despite its nutritional richness, a standardized nutrient profile is lacking. Limited genetic diversity studies underscore the need for further research. This study highlights *Mimusops zeyheri’s* multifaceted importance and research gaps, particularly in other Southern African countries. Future investigations should focus on comprehensive phytochemical analysis, ethnomedicinal validation, ecological conservation, genetic diversity assessment, and biotechnological applications. Multidisciplinary collaborations are recommended to promote sustainable utilization while preserving traditional practices.

## 1. Introduction

*Mimusops zeyheri* Sond., also known as Transvaal red milkwood, is a perennial fruit tree belonging to the Sapotaceae family. The Sapotaceae family from which *Mimusops zeyheri* belongs is more commonly referred to as the Sapodilla family and comprises an estimated 1250 species and 53 genera of tropical trees, shrubs, and branching vines [1,2,3]. Researchers have devoted considerable attention to this family on account of its ecological significance, phytochemical diversity, and economic importance [3,4,5]. Sapotaceae species contain an assortment of bioactive compounds, including triterpenoids, saponins, and phenolic compounds, according to phytochemical studies [6,7,8]. The phytochemical diversity of the Sapotaceae family was illustrated by Dasgupta, et al. [9], who determined the antioxidant and cytotoxic properties of triterpenoids extracted from *Mimusops elengi* L., an extant species belonging to the family while Lim [10] also demonstrated the phytochemical and pharmacological properties of *Chrysophyllum cainito* L., also known as star apple, with an emphasis on its possible therapeutic uses. Several studies have primarily examined the economic significance of different Sapotaceae species, specifically in relation to their contributions to the production of edible fruits, timber, and traditional remedies [1,11,12,13]. The contribution of the Sapotaceae species to biodiversity, their interactions with pollinators and seed dispersers, and their function in tropical forest ecosystems have also been the subject of ecological research [14].

*Mimusops zeyheri* is a small-to-medium-sized evergreen fruit tree reaching a maximum height of 25 m at full maturity [15]. This species is characterized by milky latex, and a well-branched, spreading, and rounded crown [16]. The bark ranges in colour from grey to dark brown with a reticulated fissure pattern [17]. The tree has an elongated trunk characterized by the absence of planks. The leaves are shiny and arranged in a spiral pattern, lacking stipules, with a petiole length ranging from 0.5 to 3.5 cm [18]. The upper leaf surface has prominent vein reticulation, while the lower surface has elevated vein reticulation. The length of the leaf blade is typically 2–3 times more than its width, and its apex is either acute to obtuse or often bluntly apiculate [18]. The lateral veins are arranged in 10–15 pairs.

The geographic distribution of this species spans across South Tropical Africa, including Angola, Malawi, Zambia, Mozambique, and Zimbabwe, extending northward to Tanzania in East Tropical Africa, and southward to Southern Africa, encompassing Botswana, South Africa, Eswatini, Lesotho Namibia [15,19]. In Egypt, *Mimusops zeyheri* heads in the direction of its variety var. *laurifolia* in terms of morphological characteristics and taxonomy, showing significant similarities and overlap that make the two taxa difficult to definitively distinguish from each other [18]. The tree is native to the northern and eastern parts of South Africa [20]. In South Africa, *Mimusops zeyheri* is primarily distributed in provinces including the Limpopo, Northwest, Gauteng, Mpumalanga, and KwaZulu Natal Provinces, and, therefore, distinctive ethnic groups associated it with distinct vernacular names, for instance, the Afrikaaner people call it “Moepel, Mmupudu (Northern Sotho), Umpushane (Zulu), Mbubululu (Venda), and Mgamba kapu in Isiswati” [15,21]. Evidence shows that *Mimusops zeyheri* typically thrives in habitats including wooded, rocky hillsides or river basins [21,22]. These species possess a non-invasive lateral root system that is well-suited for rocky habitats with poor-quality soils [23,24]. *Mimusops zeyheri* is known to exhibit optimal growth within a temperature range of 12 to 25 °C, accompanied by an average annual precipitation of 464 mm [25]. *Mimusops zeyheri*. can withstand freezing temperatures without harm and needs minimal care, including at least six hours of sunlight every day and a small amount of water to thrive [18,24,26]. This evergreen tree is drought-tolerant, salt-tolerant, and very resistant to root-knot nematodes (*Meloidogyne* species) and other pests [27].

Despite the ecological, cultural, and nutritional significance of *Mimusops zeyheri* in Africa, comprehensive information about this species remains fragmented and underutilized. The lack of synthesized knowledge on its distribution, traditional uses, phytochemical composition, genetic diversity, and potential applications hinders its conservation and sustainable exploitation. Moreover, increasing threats to its natural populations, coupled with limited scientific research, pose challenges to developing evidence-based management strategies. This study aims to consolidate existing knowledge, identify research gaps, and provide a comprehensive overview of *Mimusops zeyheri* to guide future research efforts and inform conservation strategies.

## 2. Materials and Methods

To create a thorough search string, appropriate keywords and their synonyms were identified. Using the proper truncations, alternate spellings were taken into consideration. Relevant articles for the review were found using the following search terms: *Mimusops zeyheri*, Indigenous, ethnomedicinal, nutrients, ecological roles, genetic diversity, and Biotechnological applications; these databases included Scopus, Google Scholar, EBSCO host Academic Search Complete, and ScienceDirect Once the search term was finalized, these databases were searched to find pertinent articles. The search was limited to finding data that matched the goals of the study when the “explode” option was chosen. The search was restricted to Africa, and the period of the literature search was from 1949 to April 2024. Studies that fulfilled the requirements for inclusion were downloaded to evaluate their content. Eight themes emerged and were categorized: (i) *Mimusops zeyheri*’s historical and cultural significance among Indigenous communities in South Africa; (ii) Traditional medicinal uses and healing properties associated with *Mimusops zeyheri*; (iii) *Mimusops zeyheri*’s ethnomedical uses and modes of administration; (iv) *Mimusops zeyheri* as a source of nutrients; (v) *Mimusops zeyheri*’s ecological importance and ecosystem services; (vi) Conservation status and threats to *Mimusops zeyheri* populations; (vii) Genetic diversity and population genetics in *Mimusops zeyheri*; and (viii) Scientific research and biotechnological applications of *Mimusops zeyheri*. The most thoroughly covered theme in the research questions or findings was used to categorize articles with several themes. Microsoft Word and Excel were used for the article selection, summary, and coding processes, which produced the expected minimal variability.

## 3. Literature Search Results

The systematic review protocol implemented in this study rigorously adhered to the PRISMA framework as described by Moher [28] and Shamseer, et al. [29] in Figure 1. The identification phase encompassed a multifaceted search strategy, utilizing an array of electronic databases. This primary search was augmented by reference list examination and expert recommendations, yielding an additional 11 records from dissertations, theses and books retrieved from the University of Mpumalanga (UMP) Library. The amalgamation of these search efforts culminated in a corpus of 155 documents for comprehensive full-text assessment. The subsequent screening phase entailed a meticulous examination of titles and abstracts, resulting in the exclusion of 38 documents and the retention of 117 for further scrutiny. The eligibility phase focused on the elimination of duplicate entries, necessitating the removal of 30 articles. Consequently, 87 unique documents progressed to the final inclusion stage. These 87 documents underwent rigorous full-text screening and were incorporated into the systematic review (Figure 1). Furthermore, pharmacological studies, in vitro studies, field trials, and ethnobotanical surveys reporting on the traditional use of *Mimusops zeyheri* in Africa accounted for 36.78% of the total literature in the study (Table 1). The distribution of publications on *Mimusops zeyheri* indicates a significant increase in research interest from 2016 to 2021, with 2020 and 2021 being the most prolific years, contributing 5 and 6 studies, respectively (Table 1). This surge suggests growing recognition of the species’ potential in nutrition, medicine, and agriculture, reflecting heightened global interest in indigenous and underutilized plants, sustainability, and natural health products. Notably, there were only two publications between 2002 and 2012, indicating that *Mimusops zeyheri* was under-researched during this period, possibly due to limited awareness or funding. However, starting in 2015, a steady flow of research output emerged, highlighting a focus on exploring sustainable use and integration of indigenous plants to address local challenges in sub-Saharan Africa. The concentration of recent studies suggests steadily increased awareness of *Mimusops zeyheri* as a valuable species with potential applications, which could benefit local communities economically through sustainable harvesting and cultivation practices. 

The geographical distribution of research on the species demonstrates a significant concentration in Southern Africa, with South Africa accounting for (75%) of the eligible studies, followed by Botswana and Zimbabwe (6.25%, respectively), and Eswatini, East Africa, Egypt, and Namibia (3.13%, respectively) as shown in (Table 1). The preponderance of South African studies may be attributed to the species’ widespread distribution, cultural significance, and established utilization patterns within the country [30,31]. This regional focus suggests a heightened awareness of the species potential among South African researchers compared to their counterparts in other African nations. However, this geographical disparity in research intensity may result in other Southern African countries overlooking the potential benefits of *Mimusops zeyheri*, underscoring the need for more collaborative, transnational research initiatives to comprehensively explore the plant’s potential and ensure its sustainable utilization [32,33]. Methodologically, ethnobotanical surveys predominate (46.88%) followed by in vitro studies (15.66%) and elemental analyses (12.5%), proximate analysis (6.25%), and various other approaches including field trials, feeding trials, genetic studies, taxonomic revisions, characterisation, and market surveys (3.13%, respectively). The prevalence of ethnobotanical surveys is consistent with the importance of documenting traditional knowledge and uses of the species, which serves as a crucial foundation for understanding the plant’s potential and identifying promising research directions [34]. However, the current methodological distribution suggests that research on *Mimusops zeyheri* is still in its nascent stages, emphasizing the need for more advanced laboratory techniques to elucidate the plant’s chemical composition, pharmacological properties, and potential applications across various fields [35].

**Table 1 plants-13-02943-t001:** Summary of studies on *Mimusops zeyheri* in Sub-Saharan Africa.

Reference Number	Country	Method	Major Findings
[17]	Egypt	Taxonomic revision	Provided a taxonomic revision of the *Mimusops* genus in Egypt, including *Mimusops zeyheri*, and documented its traditional uses for treating wounds and pain relief.
[18]	South Africa	Ethnobotanical survey	Identified *Mimusops zeyheri* as an important Indigenous fruit tree with suitable attributes for the semi-arid Northern Province of South Africa
[23]	South Africa	Proximate analysis	Analyzed the nutritional composition of *Mimusops zeyheri* seeds, including proteins, fatty acids, and vitamins. Reported that *Mimusops zeyheri* seeds are rich in proteins, oleic acid, and vitamin E, indicating potential as a dietary energy supplement and oil source.
[24]	South Africa	Field trial	Reported that different essential nutrient elements during and after fruiting in the soil limited the growth of *Mimusops zeyheri* trees and could be used in supplementary fertilization.
[25]	South Africa	Elemental analysis	*Mimusops zeyheri* fruits have high levels of chromium and manganese, contributing to their nutritional value.
[26]	South Africa	Characterization of endophytic fungi	Identified endophytic fungi associated with *Mimusops zeyheri* leaves, including *Teratosphaeria* and *Zeloasperium* species.
[35]	Namibia	Market survey	Identified *Mimusops zeyheri* as an important indigenous fruit tree with potential for commercialization.
[36]	South Africa	Feeding Trial	Evaluated the potential of *Mimusops zeyheri* seed meal as a substitute for maize meal in Japanese quail diets.
[37]	South Africa	In vitro assay	Reported the antiproliferative effect of *Mimusops zeyheri* seed oils on Caco-2 and HEK-293 cell lines.
[38]	South Africa	Ethnobotanical Survey	Documented the use of *Mimusops zeyheri* bark, stem, and roots for treating wounds, sores, gonorrhea, and candidiasis.
[39,40]	South Africa	Ethnobotanical survey	Surveyed and documented the use of *Mimusops zeyheri* leaves for treating tonsillitis.
[40]	South Africa	Ethnobotanical survey	Investigated the traditional uses of *Mimusops zeyheri* in Gauteng Province, including treating headaches, Hlogwana (sunken fontanelle), boosting immunity, and as a cleansing and purification agent.
[41]	Zimbabwe	Ethnobotanical survey	Reported the use of *Mimusops zeyheri* seeds as a blood purifier, and leaves for treating dysentery, boils, abscesses, convulsions, and as a sedative.
[42]	South Africa	In vitro assay	*Mimusops zeyheri* seed oil induced cytotoxic effects on MDA-MB-231 breast cancer cells and inhibited the growth of MCF-7 cells.
[43]	South Africa	Ethnobotanical survey	Documented the traditional uses of *Mimusops zeyheri* for treating snakebites, scorpion stings, arthritis, dysentery, boils, and abscesses.
[44]	East Africa	Ethnobotanical survey	Documented the use of *Mimusops zeyheri* bark for treating jaundice and as a hepatoprotective agent.
[45]	Eswatini	Ethnobotanical survey	Reported the use of *Mimusops zeyheri* leaves for treating digestive issues.
[46]	South Africa	Genetic diversity analysis	Reported 91% genetic variability among *Mimusops zeyheri* populations in Limpopo Province, South Africa.
[47]	South Africa	Ethnobotanical survey	Explored local perceptions, utilization, and population status of *Mimusops zeyheri* in the Vhembe Biosphere Reserve.
[48]	South Africa	Ethnobotanical survey	*Mimusops zeyheri* is used traditionally for food, medicine, and construction. The fruit is rich in vitamin C and has the potential for commercialization.
[49]	South Africa	Ethnobotanical survey	Reported the use of the whole *Mimusops zeyheri* plant as an aphrodisiac.
[50]	Botswana	Elemental analysis	Analyzed the presence of potentially toxic heavy metals in *Mimusops zeyheri* roots.
[50]	Botswana	Elemental analysis	Analyzed the presence of heavy metals in *Mimusops zeyheri* leaves.
[51]	Zimbabwe	Ethnobotanical survey	*Mimusops zeyheri* fruits are consumed fresh or processed into jams and jellies. The bark and roots are used in traditional medicine.
[52]	South Africa	Ethnobotanical survey	Documented the traditional use of *Mimusops zeyheri* for treating diabetes in the Vhembe District, Limpopo Province.
[53]	South Africa	In vitro assay	Leaf and bark extracts of *Mimusops zeyheri* showed antibacterial activity against various pathogens, including *Staphylococcus aureus.*
[54]	South Africa	Ethnobotanical survey	Reported the use of *Mimusops zeyheri* roots for treating weight loss.
[55]	South Africa	In vitro assay	The biological activity was reported for folkloric plants used in the treatment of ‘u wela’ including *Mimusops zeyheri*. The hexane, aqueous, and decoction extracts of *Mimusops zeyheri* showed promising antibacterial activity against *Neisseria gonorrhoeae*, with low minimum inhibitory concentration (MIC) values ranging from 0.02 to 0.03 mg/mL. The study reported the aqueous extracts of *Mimusops zeyheri* demonstrated noteworthy anti-Candida activity with an MIC value of 0.02 mg/mL against *Candida albicans*

## 4. Historical and Cultural Significance of *Mimusops zeyheri* Among Indigenous Communities in Southern Africa

*Mimusops zeyheri* is an important, underutilized tree species that local people associate with, providing various ecosystem services, including fruits, phytomedicine, cooking oil, and timber [47]. According to Omotayo, et al. [56] *M. zeyheri* holds a significant role and it is largely used by local people in the Southern Africa region. From this context, it is arguable that this species serves as a historical emblem of cultural and ecological importance in South Africa. *Mimusops zeyheri* is not merely a botanical entity in Indigenous communities; it is also deeply ingrained in traditional customs, beliefs, and folklore [26,47]. The tree’s strong and tough reddish-brown timber has been highly valued throughout history for its adaptability and was thus used in the creation of traditional equipment (wooden axe), elements and devices of folk music and ceremonial artefacts [57]. The tree’s solid wood which has been believed to possess metaphorical significance commonly used in rites and ceremonies that represent strength, endurance, and ancestral ties [58]. In several indigenous cultures, the tree holds profound significance, deeply intertwined with myths, legends, and spiritual beliefs (Table 2). It is regarded with utmost reverence as a sacred entity embodying ancestral wisdom and providing protection to those who utilize it according to ancestral guidance [45]. The crucial role of this indigenous wild fruit tree in cultural ceremonies, rites of passage, and spiritual rituals like *Inthwaso* (a Zulu spiritual initiation) underscores its embodiment of cultural continuity and identity among indigenous communities [18]. The tree serves as a living connection to heritage, traditions, and the collective ancestral knowledge passed down through generations [15,18].

## 5. Traditional Medicinal Uses and Healing Properties Associated with *Mimusops zeyheri* (Table 2)

Traditional medicine has a long and storied history, with most of the world’s medicinal plant populations originating from Asia, Africa, and Latin America [71]. Ojewole [72] estimates that more than 80% of Africans still use traditional medicines made from plants for both personal care and medicinal purposes. According to Van Wyk [23], approximately 25% of the world’s higher plants, including wild fruit trees are found in Southern Africa. This makes the Southern African region fall amongst the richest plant diversity regions globally. Other species within the genus *Mimusops* including *Mimusops elengi* Linn., and *Mimusops hexandra* (Roxb.), also possess a range of pharmacological properties, including antibacterial, anti-inflammatory, and antiulcer effects [59,60]. Literature evidence informs that most plant species that share similar genera possess similar medicinal properties [55]. Undoubtedly and arguably so, *Mimusops zeyheri.* could possess similar medicinal properties as other species that belonged to a similar genus. Yet recent scientific evidence revealed that there is still a dearth of studies that have specified the medicinal usage of *Mimusops zeyheri* in the Southern African context [40].

### 5.1. Plant Part Used

This study reported the use of nine distinct plant parts of *Mimusops zeyheri* in Africa for the treatment of various ailments. Based on the critical analysis of (Table 2), leaves (34%) emerged as the most used plant part of *Mimusops zeyheri* cultivated for ethnomedicinal uses followed by roots (32%) as shown also in (Figure 2). The predominant use of the two plant parts aligns well with African traditional medicine and can be attributed to several factors including its relative availability all year round, ease of collection or ethnic beliefs of local people [49,73,74]. In addition, Alamgeer, et al. [75] suggest that the preference for leaves over other plant parts may be attributed to their role as photosynthetic organs, containing photosynthates that may contribute to medicinal properties. Roots are preferred as they are regarded traditionally as they are believed to possess more pharmacological properties [49]. In some cultures, the roots are mixed with a spider’s web, which is pounded and taken orally with warm water thrice a day (Table 1) for the treatment of tuberculosis. Although regarded as strong, the collection of underground plant parts is not viable, as it has a detrimental impact on the plant’s existence and can ultimately be regarded as highly endangering the plant [61]. According to Mahomoodally [64], the preference of both leaves and roots for ethno-preparations is attributed to their high concentrations of pharmacologically active substances. The significant use of bark (13%) suggests a perceived efficacy, likely due to its concentration of bioactive compounds though this raises sustainability concerns due to potential overexploitation [65,76]. Other plant parts, including whole plant (6%), bark and roots combined (4%), and seeds (2%), show moderate usage levels. Lower utilization is noted for stem and bark (8%), fruits and flowers (2%). This diverse usage pattern suggests a comprehensive understanding of *Mimusops zeyheri’s* potential benefits among local people and its contribution to livelihood.

### 5.2. Methods of Preparation

The preparation methods are of paramount importance in ethnomedicine. Preparation methods vary significantly based on the specific plant species, plant parts utilized, geographical location, and cultural context [66,74]. In this study, the most common preparation method is infusion, accounting for 33% of all methods (Figure 3). The predominance of infusion as a preparatory method aligns with findings from various ethnobotanical studies conducted across diverse African ethnic groups [62,63,77]. The prevalence of this method can be attributed to its simplicity and efficacy in extracting bioactive compounds from plant materials. The ease of implementation and the generally high bioavailability of water-soluble phytochemicals contribute to the method’s popularity [71]. Decoction, closely following at 31%, represents another frequently employed technique. The comparable utilization rates of infusion and decoction are noteworthy, as both methods involve aqueous extraction of plant constituents, though, under different thermal conditions and extraction durations. This predilection for water-based extraction methodologies may be ascribed to their accessibility and deep-rooted cultural significance in traditional therapeutic practices [78]. Moreover, the preference for these methods may be partly explained by the thermostability of certain phytochemicals and the enhanced extraction efficiency at elevated temperatures [71]. Powder preparation, constituting 19% of all procedures, underscores the significant role of desiccated plant material in traditional medicinal applications. This technique offers advantages in terms of extended shelf-life and versatility in administration, which can be particularly beneficial in resource-limited areas [79]. Additionally, the process of pulverization can increase the surface area of plant material, potentially enhancing the extraction of bioactive compounds during subsequent preparation or administration [80]. The remaining methodologies comprise pounded paste (8%), tincture (3%), and other unspecified processes (3%). The relatively low prevalence of tinctures is noteworthy, as this method typically involves alcohol-based extraction. This lower representation may be attributed to cultural preferences, resource limitations, or concerns regarding the use of alcohol in certain communities [81]. Furthermore, the efficacy of alcohol-based extractions in preserving certain thermolabile compounds may not outweigh the sociocultural barriers to its widespread adoption in traditional medicine practices [82].

### 5.3. Administration Method

The distribution of administration methods for *Mimusops zeyheri* in ethnomedicine indicates a high preference for oral administration (70%) in (Figure 4). This usage pattern is consistent with several ethnobotanical research on Sapotaceae and other medicinal plants, demonstrating *Mimusops zeyheri’s* adaptability in traditional medicine [83]. The prevalence of oral administration indicates that *Mimusops zeyheri* may contain bioactive substances that are efficacious when consumed. This is consistent with studies from other Sapotaceae species, such as *Mimusops elengi*, which has demonstrated considerable pharmacological activity when taken orally [83]. *Mimusops zeyheri’s* high rate of oral use could be due to the existence of comparable bioactive chemicals that are well absorbed through the gastrointestinal system which could potentially heal a variety of internal disorders [84]. Topical application (12%) and poultice use (10%) make up a sizable amount of *Mimusops zeyheri’s* traditional use, demonstrating its potential to heal skin conditions and exterior ailments. Shekhawat, et al. [85] reported that topical application of plant extracts had considerable wound-healing effects, which were attributed to their antibacterial and anti-inflammatory actions. The usage of *Mimusops zeyheri* in topical and poultice forms implies that it may have similar qualities, making it useful for treating skin infections, wounds, and inflammatory skin conditions [86]. The inhalation method (6%), although uncommon, is an intriguing component of *Mimusops zeyheri’s* traditional use. Inhalation as a method of delivery is frequently related to the treatment of respiratory problems or its aromatherapeutic characteristics [86]. While studies on inhaled *Mimusops zeyheri* formulations are few, research on other Sapotaceae species sheds light on potential benefits. In vitro testing of essential oils derived from *Manilkara zapota* (L.) P.Royen leaves showed antibacterial activity against respiratory infections [87]. *Mimusops zeyheri’s* inhalation may be comparable, with its volatile chemicals having good effects on the respiratory system or providing relief through aromatherapy [88].

### 5.4. Ethnobotanical Uses of Mimusops zeyheri

Traditional medicine is important in treating many illnesses, especially in rural African areas. *Mimusops zeyheri* has been utilised for generations and is an essential component of healthcare systems in many regions [61]. The ethnomedicinal use of *Mimusops zeyheri* in Africa shows a wide spectrum of medicinal and therapeutic applications across several physiological systems (Figure 5). In this study, skin conditions (18%) are the most reported treated conditions, followed by infections and immune system-related issues (16%). This is consistent with findings from earlier ethnobotanical research, which frequently highlight dermatological and immune-related uses of *Mimusops zeyheri* [23,61]. The plant’s putative antibacterial and anti-inflammatory qualities, which are found in many medicinal herbs used topically, may explain the plant’s high prevalence in the treatment of skin conditions [89]. The significant prevalence of treated infections and immune system conditions (approximately 17%) suggests that *Mimusops zeyheri* may contain bioactive substances with immunomodulatory properties, necessitating further pharmacological research [67]. Metabolic and organ conditions, as well as other undefined illnesses, account for approximately 13–14% of all ailments reported to be treated with *Mimusops zeyheri*. This wide range of applications demonstrates the plant’s adaptability in traditional medicine systems and may reflect its ability to influence different physiological pathways [90]. Pain alleviation, reproductive health, fertility, and respiratory difficulties account for roughly 11–12% of recorded uses. The analgesic capabilities suggested by its usage in pain management could be linked to anti-inflammatory chemicals, whilst its use in reproductive health could indicate hormonal or fertility-enhancing activities [91]. Rural African people frequently choose traditional herbs such as *Mimusops zeyheri* for a variety of reasons. These include accessibility, as medicinal plants are frequently available in local settings, making them more accessible than modern medications in rural locations [64]. Another issue to consider is cultural relevance, as traditional medicine is strongly ingrained in local cultures and belief systems, making it more accepted and trustworthy. Affordability is also important, as picking native flora is typically more cost-effective than acquiring manufactured medications, particularly in resource-constrained contexts [23]. However, long-term use generates vital research leads but does not ensure safety or efficacy. Street, et al. [92] emphasise the importance of comprehensive scientific testing of traditional medicines to determine their pharmacological characteristics and potential toxicity. The preference for indigenous herbs in rural African communities, despite the arrival of modern treatment, emphasises the importance of adopting an integrative healthcare system.

## 6. *Mimusops zeyheri* as a Source of Nutrients

Wild fruits are natural storehouses of essential macro and micronutrients [93]. *Mimusops zeyheri* is a valuable fruit tree with a pool of significant nutrients essential to the human body. According to Chivandi, et al. [22], *Mimusops zeyheri* seeds contain high levels of proteins, oleic acid and vitamin E indicating its potential for effective use as a dietary energy supplement and oil source. A notable vitamin E content ranging between 0.50 and 48.7 µg was reported by Chivandi, et al. [36]. Gomes, et al. [42] reported the highest levels of linoleic acid, measuring 18.87% from the seed meal of *Mimusops zeyheri*. Lipids are crucial to the physiological processes of plant growth, and they have a close association with human metabolism, among other functions [94]. Gomes, et al. [42] reported a 14.04% lipid yield from the seed oils of *Mimusops zeyheri* and the highest total soluble fatty acid content of 23.902% compared to *Kigelia africana* (Lam.) Benth. and *Ximenia caffra* Sond. which yielded 17.15% and 8.57%, respectively. The proximate analysis study of the seeds of *Mimusops zeyheri* by Chivandi, et al. [22] reported a lipid yield of 21.2% with a gross energy value of 24.34 MJ/kg. The seed oil of *Mimusops zeyheri* has the highest reported concentration of glutamic acid in comparison to other amino acids accounting for 1.38% of crude protein found in total fat [95]. Dehulled seed meal from *Mimusops zeyheri* can substitute a portion of maize meal in Japanese quail diets to provide energy without affecting growth, feed utilisation efficiency, or meat output [96]. At 37.5% diet inclusion, this substitution is economically viable [96]. The use of *Mimusops zeyheri* seed meal may help produce and increase the output of Japanese quail meat. Although there have been studies on the use of *Mimusops zeyheri* seed meal as a feed supplement for quail and poultry, there is a dearth of studies on its applications in other fields such as the food industry, nutraceuticals, or medicines. There is, therefore, a need for preliminary investigations to discover and assess the uses of *Mimusops zeyheri* in many sectors, such as food fortification, functional foods, organic preservatives, or medicinal substances, leveraging its distinctive nutritional composition and bioactive components.

Venter and Venter [21] reported plant fresh fruit concentrations ranging between 50 and 80 mg/100 g. The dietary benefits of the fruits derived from this tree are notable, mostly attributed to their higher levels of chromium and manganese [26]. The fruits of *Mimusops zeyheri* were reported to have a vitamin C content higher than that of guava vitamin C content of approximately 0.20 mg/kg^−1^ [56] and oranges [18,56], suggesting that the fruits can be a valuable addition to the diets of rural people who may especially struggle with limited access to fresh produce containing vitamin C. Wilson, et al. [97] reported higher levels of carbohydrates (fructose, glucose and sucrose) from the fruits of *Mimusops zeyheri*. These serve as a good energy source and would aid in the active functioning of rural people and school children who might otherwise lack an alternative source of energy. The fruits *of Mimusops zeyheri* exhibit higher levels of dry matter organic matter, protein, and ash content, precisely measuring at 91.1%, 83.3%, 9.3%, and 2.8%, respectively, as reported in a study conducted by Chivandi, et al. [36].

Okatch, et al. [50] determined the composition of *Mimusops zeyheri* leaves which included nitrogen (N) at a concentration of 6.33%, phosphorus (P) at 0.33%, potassium (K) at 1.25%, calcium (Ca) at 0.39%, magnesium (Mg) at 0.06%, zinc (Zn) at 0.0029%, copper (Cu) at 0.0014%, iron (Fe) at 0.0409%, aluminium (Al) at 0.007407% and manganese (Mn) at about 0.005185%. Another research that examined the root of *Mimusops zeyheri* in Botswana discovered trace amounts of chromium (Cr) arsenic (As) and lead (Pb) measuring approximately 0. 73 mg/kg^−1^, 1.73 mg/kg^−1^ and 0.20 mg/kg^−1^, respectively [50]. All these constituents were found to fall within limits thus confirming the plants’ suitability, for medical and dietary purposes. Despite various research examining individual nutrients in *Mimusops zeyheri*, there is still a lack of a comprehensive analysis of the plant’s complete nutrient profile, which includes fruits, seeds, leaves, and other components such as the roots. This could offer a more comprehensive comprehension of the plant’s nutritional worth and its uses. There is, therefore, a need to utilize modern analytical techniques to perform a comprehensive and standardized nutrient analysis of every part of the *Mimusops zeyheri* plant, encompassing macro and micronutrients, phytochemicals, and bioactive substances. Although the nutrient composition of *Mimusops zeyheri* has been documented, information is scarce regarding the bioavailability and bio-accessibility of these nutrients. This knowledge is essential for comprehending the possible health advantages and efficient utilization of these nutrients by the human body.

## 7. Ecological Importance and Ecosystem Services Provided by *Mimusops zeyheri*

Wild fruit trees are essential for maintaining a balanced and functional ecosystem. Several wild fruits especially those from the Sapotaceae family including *Mimusops zeyheri* are host to several endophytes [26] and maintain a symbiotic relationship with them [98]. *Mimusops zeyheri* has a mutualistic relationship with *Teratosphaeria*, *Zeloasperium* species, *Pezizomycotina* endophytes [26]. These endophytes live inside *Mimusops zeyheri* acquiring good nutrition from the fragments, exudates and leachates of the tree without causing harm. In return for what they acquire from the fruit tree, the endophytes provide a physiological impact on the tree by increasing its resistance to environmental and biological stresses [50]. This relationship serves as an integral component of forest dynamics and influences greatly the survival and regeneration of wild fruit trees [98].

In addition to being a host for endophytes, *Mimusops zeyheri* serves also as a key habitat provider, food source, and soil stabilizer [99,100,101]. Its dense foliage and substantial canopy create a suitable microclimate for various birds and insect species, while its fruits are an important food source for many animals [99]. Although the tree offers these services, there is a lack of information regarding its extensive interaction with other biotic elements of the environment, including insects, birds, and other animal species. A potential approach is to conduct extensive ecological research and biodiversity surveys to gain insight into the complex network of relationships that exist between *Mimusops zeyheri* and different animal species, such as herbivores, pollinators, and seed dispersers. This may shed light on the tree’s function in preserving ecosystem balance and encouraging biodiversity.

The tree’s extensive root system helps prevent soil erosion, and its fallen leaves contribute to nutrient cycling [98]. However, the tree’s role in competition and succession within its ecosystem is not well understood [99]. Conducting extensive ecological studies and field observations to gain insights into the competitive dynamics of *Mimusops zeyheri* with other plant species, as well as its contribution to successional processes within the ecosystem. This might involve maintaining close monitoring of any changes to the composition of the vegetation, researching the use of resources, and assessing *Mimusops zeyheri*’s effects on other species.

According to England, et al. [102], *Mimusops zeyheri* is essential for maintaining ecosystem services and natural capital in grazed ecosystems. However, it is unknown how specifically *Mimusops zeyheri* contributes to ecosystem services such as soil stabilization, carbon sequestration, and water resource regulation. Potential resolution: Perform field studies and experimental setups to measure the ecosystem services offered by *Mimusops zeyheri.* This could include the measurement of carbon sequestration rates, the evaluation of soil erosion rates in areas with and without *Mimusops zeyheri,* and the examination of the tree’s influence on water cycles and hydrological processes.

With its considerable nutritional and commercial potential, *Mimusops zeyheri* is a valuable resource for food security and sustainability [56].Although Syampungani, et al. [103] reported *Mimusops zeyheri* to play a crucial role in climate change mitigation and adaptation within the Miombo woodlands, it remains unclear and poorly known how *Mimusops zeyheri* specifically contributes to climate change adaptation and mitigation within its environment. Hence, it is imperative to research to assess the function of *Mimusops zeyheri* in controlling microclimate conditions, regulating temperature and moisture levels, and its capacity to serve as a safeguard against severe weather catastrophes. This may entail doing on-site measurements, utilizing remote sensing methods, and employing ecological modelling instruments. Beyond some of the potential ecosystem services provided by the plant species in the ecosystem, *Mimusops zeyheri* offers a range of ecosystem services that are crucial for rural people, like fuelwood, fodder, and medicinal plants [102].

## 8. Conservation Status of *Mimusops zeyheri* Populations

In South Africa, the South African Biodiversity Institute (SANBI) is responsible for evaluating the conservation status of plant species including wild fruit trees in the country [104,105]. Assessments by the SANBI are crucial in the guide for conservation and they are submitted to the International Union of Conservation of Nature (IUCN). *Mimusops zeyheri* is, at present, categorized under the “Least Concern” list of indigenous wild fruit trees by the IUCN [105]. This categorization does not imply that this wild fruit tree species is not facing any threats. The tree has been documented as a host to the Mediterranean fruit fly which was reported to pose significant harm and threat to the horticulture industry of South Africa [106,107,108]. Although the species is currently not in any immediate danger of extinction, its distribution and survival are confronted by several threats. The results of the study by Lubisi, et al. [47] in the Vhembe biosphere reserve highlight that 54.14% of the participants observed population changes in the tree and noted that the species is steadily declining, scattered in distant areas and around valleys. Local rural people in Africa rely on the exploitation of wild fruits for food, medicine, firewood and timber often used for handicrafts. The exploitation of these wild fruit trees often gets excessive and alongside factors such as climate change, the introduction of alien invasive species, expansion of agricultural land and residential areas, continuous urbanisation poses a great threat to their existence [109,110,111]. The effects of such threats, more especially urbanisation, are evident in the study by Matlala, et al. [40], who reported a declining and scarce occurrence of *Mimusops zeyheri* as mentioned by residents in the Gauteng Province of South Africa often referred to as the country’s most urban province.

Historically, local farmers in many African countries such as Botswana [50], Zimbabwe [41] and Eswatini [45] carried out the protection of wild fruit trees and plant resources around their forest reserves and farm homelands [93]. South Africa has established protected areas including national parks, botanical gardens and nature reserves to help conserve biodiversity including wild fruit trees. These areas operate under strict national regulations including acts such as Conservation of Agricultural Resources Act No. 43 of 1983, National Forests Act No. 84 of 1998, National Environmental Management Act No. 107 of 1999, and National Environmental Management: Biodiversity Act No. 10 of 2004 as mentioned several scholars [111,112,113,114]. Botanical gardens and arboreta nurseries play a vital role in ex situ conservation initiatives, especially for ancient trees such as *Mimusops zeyheri* [115]. Even though these establishments are deemed useful and progressive, Scientific research argues that they prevent local people from using their surrounding biological resources and receiving direct subsistence advantages [116]. Furthermore, there are obstacles to overcome with these establishments, such as the absence of a well-established scientific and legal structure for ex situ operations [117], and the requirement for a synchronized meta-collection infrastructure to monitor and administer living collections [118]. To successfully tackle these issues, Volis [119] proposes the establishment of regional conservation priorities, the creation of genetically diverse collections, and the utilisation of these collections in on-site initiatives. These measures can improve the function of botanical gardens and arboreta in ex situ conservation.

## 9. Genetic Diversity and Population Genetics in *Mimusops zeyheri*

The genetic and population diversity of various wild fruit tree species have been documented across the globe; however, such documentation in South Africa and the African continent remains scanty. Significant genetic structures isolated by distance were reported in certain wild fruit tree populations [120]. Research on genetic variability is important in the survival and evolutionary potential of the species [121] as it may provide valuable insights into the identification and classification of the plant within breeding programmes hereby providing recommendations for protection and conservation measures [56]. The genetic variability of *Mimusops zeyheri* in South Africa has only been documented in the Limpopo Province. A study by Ledwaba [46] reported 91% genetic variability among *Mimusops zeyheri* populations with 9% population variability. This suggests a research gap exists and, therefore, prompts a question regarding the presence of comparable genetic patterns or the presence of distinct genetic profiles in other provinces of the country where *Mimusops zeyheri* exists. In the Limpopo Province, *Mimusops zeyheri* is divided into several different clusters with an average genetic similarity estimate that is between 47 and 89% [46] suggesting different levels of genetic relatedness. Further investigation regarding factors that contribute to the formation of such genetic clusters could lead to valuable knowledge generation regarding the historical patterns of tree migration, environmental influences and possible bottlenecks to gene flow within the wild fruit tree species. Such knowledge will also help in identifying populations to prioritize for conservation efforts and the development of corridors for restoration of the plant species and its habitat.

## 10. Scientific Research, Phytochemistry and Biotechnological Applications of *Mimusops zeyheri*

Although *Mimusops zeyheri* is highly valued as an important traditional wild fruit tree crucial for its medicinal and miscellaneous uses in rural livelihoods, reports of the plant’s biological activities are scanty. The first biological activity associated with *Mimusops zeyheri* was reported by Chivandi, et al. [36], who reported the antiproliferative effect of *Mimusops zeyheri* seed oils on Caco-2 and HEK-293 cell lines. These results indicate a promising avenue for further research on the seed oil mechanism of action and potential application. In a similar study by Gomes, et al. [42], *Mimusops zeyheri* seed oil reportedly induced cytotoxic effects on MDA-MB-231 cells and growth-inhibitory effects on MCF-7 cells. Additional research is required on the in vitro biological activities of different parts of the plant extracts against various human diseases to scientifically support their use in ethnomedicine. These studies may warrant a potential for the inclusion of the plant in the pharmaceutical industry. Research aimed at exploring the phytochemical constituents of the whole plant is also required. This will create a space for the potential development of novel drugs for the treatment of various health conditions and applications in the cosmetic industry for the formulations of many beauty products such as moisturizers for treating skin conditions. Additional investigation is required to examine the possible biotechnological uses of *Mimusops zeyheri*, in the advancement of functional foods nutrition-derived products.

## 11. Commercialization of *Mimusops zeyheri*

Despite its significant nutritional, medicinal, and ecological potential, *Mimusops zeyheri* remains underutilized, facing several limitations that hinder its extensive use and commercialization. One of the primary constraints is the fragmented scientific knowledge of the species. While *Mimusops zeyheri* is acknowledged for its traditional uses, detailed studies on its phytochemistry, pharmacological properties, and comprehensive nutritional profile are limited [56]. The lack of standardized research, particularly in underexplored regions of Southern Africa, constrains its development as a commercial product [40].

A key challenge for commercialization lies in the sustainability of harvesting practices. In many rural communities, wild-harvesting is the predominant method for obtaining *Mimusops zeyheri* products, which can lead to overexploitation and environmental degradation [56]. Similar to other wild fruit species, overharvesting of key plant parts such as roots, bark, and seeds can threaten the long-term viability of *Mimusops zeyheri* populations [94]. This challenge is exacerbated by increasing urbanization and agricultural expansion, which result in habitat loss and fragmentation, reducing the natural range of the species [40].

Limited market infrastructure and value chain development for wild fruits, including *Mimusops zeyheri*, pose a significant barrier to its commercialization. Smallholder farmers and harvesters often lack access to organized markets, which restricts the plant’s economic potential [122]. In addition, the absence of processing facilities for value addition, such as the extraction of oils, vitamins, and other bioactive compounds from *Mimusops zeyheri*, limits its appeal in both local and international markets [24].

*Mimusops zeyheri* is highly tolerant to climate extremes such as droughts and predatory nematodes; however, the lack of genetic improvement and cultivation techniques makes extensive adoption and use of the plant challenging. Unlike other commercially successful fruit species, *Mimusops zeyheri* has not undergone selective breeding or genetic improvement programs aimed at enhancing its yield, disease resistance, or adaptability to varying agroecological zones [46]. This results in variable fruit quality and inconsistent supply, further impeding its commercialization. This is compounded by the species’ slow maturation process, which discourages smallholder farmers from investing in its domestication due to delayed economic returns [122]. Developing effective agronomic and propagation practices that can reduce the time to fruiting and enhance yield will be key to its commercialization.

The regulatory framework governing the commercialization of indigenous wild fruit trees in Southern Africa is underdeveloped. Current regulations do not provide adequate support for the sustainable utilization and commercialization of species like *Mimusops zeyheri* [122]. Without clear guidelines on sustainable harvesting, intellectual property rights, and benefit-sharing, there is a risk of both resource depletion and unfair exploitation of indigenous knowledge about the plant and resources [46].

## 12. Research Gaps

This study provides details on the nutritional composition from various studies [25,42] a comprehensive and standardized analysis across all plant parts is lacking. As Omotayo, et al. [56] emphasize, understanding the complete nutrient profile and bioactive compounds is crucial for leveraging the plant’s potential. However, the study also notes the scarcity of information on the bioavailability and bio-accessibility of these nutrients, which Ramarumo, et al. [111] argue is essential for understanding their efficient utilization and health impacts. This study, therefore, recommends the utilization of modern analytical techniques to perform a comprehensive nutrient profiling of every part of the *Mimusops zeyheri* plant, encompassing macro and micronutrients, phytochemicals, and bioactive substances. Additionally, co-studies to assess the bioavailability and bio-accessibility of these nutrients to better understand their potential health impacts and optimal utilization are necessary.

The lack of clinical studies and potential therapeutic applications is an issue to be addressed around *Mimusops zeyheri*. Several studies primarily focus on the nutritional composition of the plant, with limited information on potential therapeutic applications or clinical studies [25,96]. Conducting clinical studies could provide valuable insights into the potential medicinal or therapeutic benefits of the plant or its components. Therefore, there is a need to carry out clinical trials and preclinical studies. Designing and conducting preclinical studies to evaluate the potential therapeutic effects of the plant or its extracts on various health conditions or disease models can bridge the existing gap. Based on the preclinical findings, conducting clinical trials to assess the safety, efficacy, and potential therapeutic applications of the plant or its components in human subjects can provide valuable insights into the planned use and its application. There is also a need to explore traditional knowledge and ethnobotanical data by collaborating with local communities and traditional healers to gather information on the traditional uses and preparation methods of the plant. This traditional knowledge can be Integrated with scientific research to identify potential therapeutic applications and develop evidence-based products or treatments.

Authors like Kozlowski, et al. [115] and Ramarumo, et al. [111] emphasize that conservation and utilization are inextricable, while Omotayo, et al. [56] caution against unchecked utilisation which may lead to environmental degradation. As argue, that sustainable decision-making requires integrating multidisciplinary knowledge, including indigenous knowledge systems. We need to foster collaborations between researchers, institutions, stakeholders (including local communities), and traditional knowledge holders to facilitate knowledge exchange, resource sharing, and integrated approaches. Multidisciplinary collaborations can effectively address gaps and promote sustainable utilization while preserving traditional practices.

While Majeed, et al. [123] reported on the biological activities of the *Mimusops* genus, and the study mentions the potential for biotechnological applications and inclusion in pharmaceuticals, there is a lack of specific research in this area for *Mimusops zeyheri*. This, therefore, means there is a need for studies aimed at the comprehensive phytochemical analysis of the whole plant and investigation of potential biotechnological uses, including developing functional foods, nutraceuticals, cosmetics, and novel drug candidates, capitalizing on the plant’s unique composition and bioactive compounds. Addressing these gaps through future research can contribute to a comprehensive understanding of *Mimusops zeyheri*’s potential benefits, sustainable utilization, and conservation strategies while fostering collaborations between scientific and traditional knowledge systems.

## 13. Conclusions

*Mimusops zeyheri* emerges as a multifaceted indigenous plant species with significant ecological, cultural, and potential economic value. The extensive ethnomedicinal applications documented in some African countries underscore its importance in traditional healthcare systems. The diverse usage of different plant parts for treating a wide range of ailments suggests a rich phytochemical profile worthy of further scientific investigation. The nutritional analysis of *Mimusops zeyheri* fruits and seeds reveals a promising source of essential nutrients, particularly vitamins and minerals, which could contribute to food security and nutrition in rural communities. However, the lack of a standardized nutrient profile across all plant parts indicates a critical area for future research. Ecologically, *Mimusops zeyheri* plays a crucial role in its native habitats, supporting biodiversity and providing essential ecosystem services, with its endophytic interactions and potential contributions to soil stabilization and climate change mitigation warranting further investigation. Despite its current “Least Concern” conservation status, the species faces threats from overexploitation, habitat loss, and climate change, necessitating more robust, integrated conservation strategies. Limited genetic diversity studies suggest significant variability within populations, highlighting opportunities for targeted conservation and varietal improvement, though substantial knowledge gaps persist across the species’ range. Preliminary biological studies have revealed promising cytotoxic and antiproliferative activities in *Mimusops zeyheri* seed oils, opening avenues for potential pharmaceutical applications. However, the scarcity of comprehensive phytochemical analyses and clinical studies underscores significant research opportunities in exploring the full potential of this valuable indigenous species. Notably, this study has highlighted a significant disparity in research in some African countries, with most studies concentrated in South Africa. This geographical imbalance in research intensity may result in overlooking the potential benefits and unique characteristics of *Mimusops zeyheri* populations in other Southern African nations. The limited research in most African countries where *Mimusops zeyheri* is native represents a critical gap in our understanding of its full potential and variability across its natural range. To address the research imbalance, the establishment of collaborative, transnational research initiatives focused on *Mimusops zeyheri* is necessary. Such collaborations should aim to comprehensively explore the plant’s potential across its entire natural range, ensuring a more holistic understanding of its variability, uses, and conservation needs. This approach would not only enhance our scientific knowledge but also promote equitable development and sustainable utilization of this valuable resource across all relevant African nations.

## Figures and Tables

**Figure 1 plants-13-02943-f001:**
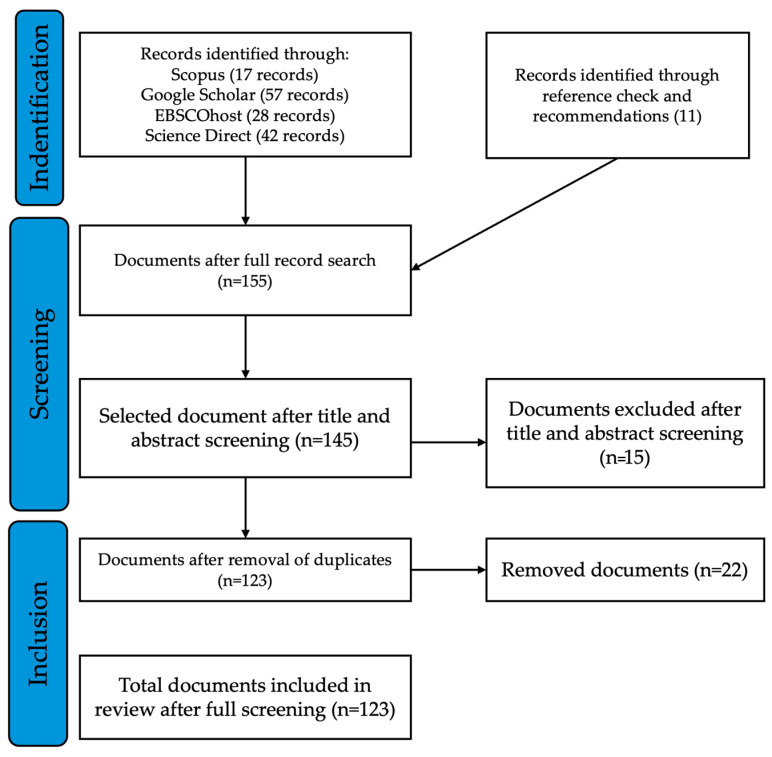
The literature search method used for the selection of articles included in this review.

**Figure 2 plants-13-02943-f002:**
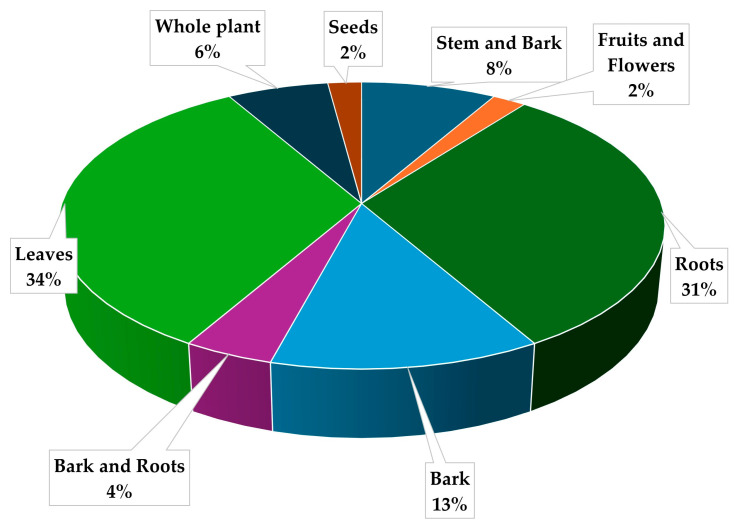
*Mimusops zeyheri* plant parts used for managing different ailments and conditions in Southern Africa.

**Figure 3 plants-13-02943-f003:**
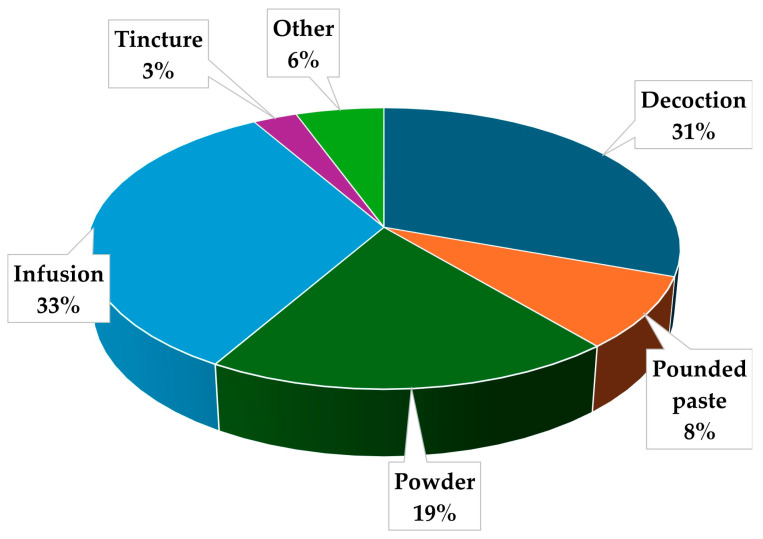
Methods of preparing *Mimusops zeyheri* for treatment of different ailments and conditions in Southern Africa.

**Figure 4 plants-13-02943-f004:**
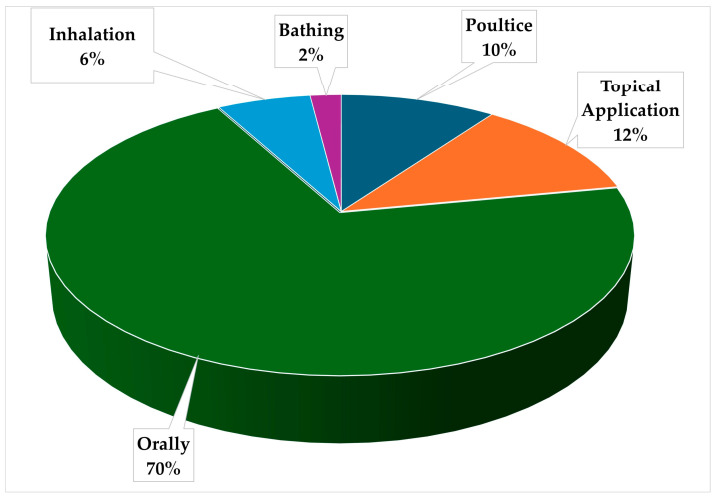
The administration method of ethnomedicine from *Mimusops zeyheri* is used to treat different ailments and conditions in Southern Africa.

**Figure 5 plants-13-02943-f005:**
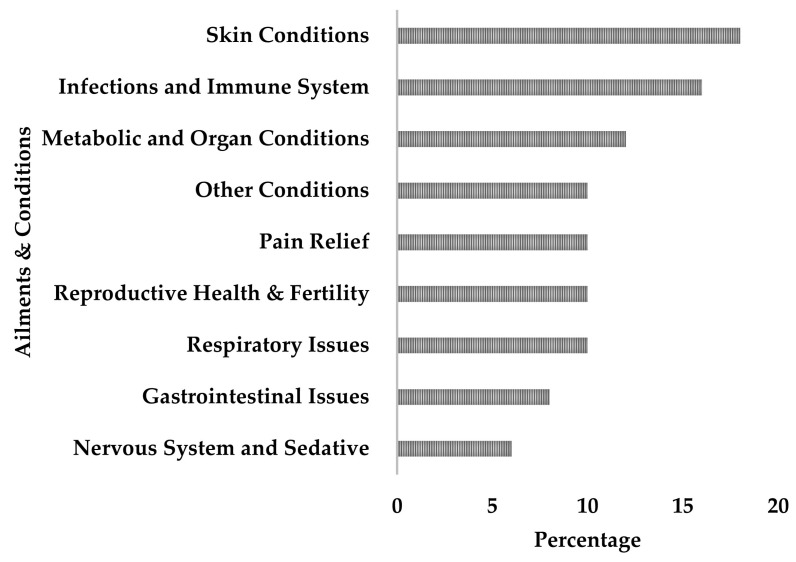
Categories of ailments and conditions treated with *Mimusops zeyheri* in Southern Africa.

**Table 2 plants-13-02943-t002:** Ethnomedicinal uses of *Mimusops zeyheri* and modes of administration in Southern Africa.

Ethnobotanical Use	Plant Part	Method of Preparation	Administration	References
Treatment of wounds	Stem and bark	The bark and stem are ponded together to create a paste with some oil added to it.	Topical application	[18]
Pain relief	Stem and Bark	Tincture	Orally	[18]
Treatment of stomachache	Bark	Decoction	Orally	[25]
Treatment of wounds and sores	Stem and Bark	Decoction and pulverization (Bark is dried and crushed into fine powder)	Poultice.	[25,37]
Treatment of wounds (Tilondza)	Fruits and Flowers	Pounded into paste	Poultice	[25,53]
Treatment of gonorrhea	Roots	Infusion and Decoction	Orally	[38]
Treatment of candidiasis	Bark	Infusion and Decoction	Orally	[38]
Treatment of tuberculosis	Roots	Infusion and Decoction	Orally	
Treatment of womb issues	Roots	Infusion and Decoction	Orally	[38,53]
Tonsils	Leaves	Infusion	Orally Gagle with the leave infusion to relief pain from tonsils	[41]
Blood purifier	Seeds	Powder and Infusion	Orally	[42]
Treatment of diabetes	Seeds	Powder and Infusion	Orally	[43]
Arthritis	Leaves	Pounded into paste	Topical application	[44]
Dysentery	Leaves	Infusion and decoction	Orally	[42,44]
Treatment of boils and abscesses	Leaves	Pounded into paste	Poultice	[42,44]
Convulsions	Roots	Infusion and decoction	Orally	[42]
Sedative	Leaves	Infusion and decoction	Orally	[42]
Snake bite	Leaves	Infusion	Topical application	[44]
Scorpion stings	Leaves	Infusion	Topical application	[44]
Treatment of diabetes	Seeds	Powder and Infusion	Orally	[44]
Arthritis	Leaves	Pounded into paste	Topical application	[44]
Jaundice	Bark	Infusion	Orally	[45]
Used as hepatoprotective	Bark	Decoction	Orally	[45]
Digestive issues	Leaves	Powder	Orally	[46]
Treatment of sunken fontanelle in infants	Leaves	Leaves are dried and later burnt to create ashy powder	Poultice	[53]
Boosting immunity in humans	Roots	Infusion	Orally	
Ethnoveterinary, to improve livestock sexual performance (goats) during the breeding season	Leaves	Leaves are mixed with animal feed	Orally	[53]
Treatment of headache	Roots	Roots are dried and pulverized into fine powder	Snorted like snuff and inhaled as smoke when burned	[53]
Treatment of weight loss	Roots	Decoction	Orally	[53]
Treatment of menstrual pains	Leaves	Infusion	Orally	[59]
Aphrodisiac	Whole plant (Leaves, barks, and seeds)	Decoction, infusion, or powder	Orally	[60,61]
Treatment of erectile dysfunction	Whole plant (Leaves, barks, seeds, roots)	Leaf infusion, Seeds are dried and pulverized to mix powder with food. Root decoction	Orally	[62,63]
Cleansing and purification	Bark and roots	Infusion, Decoction	Orally and bathing	[64]
Treatment of ulcers	Bark and roots	Decoction	Orally	[65]
Treatment of foul-smelling discharge	Leaves	Infusion	Orally	[66]
Treatment of gastrointestinal issues	Whole plant (Leaves, barks, and seeds)	Decoction	Orally	[66]
Treatment of *diabetes mellitus*	Leaves	Decoction	Orally	[66]
Tuberculosis	Roots	Infusion Mixed with (spider’s web). Pounded and taken orally with warm water. Thrice a day	Orally	[67]
Fever reduction	Roots	Powder	Orally	[67]
Treatment of gum inflammation	Roots	Decoction	Orally	[67]
Treatment of toothache	Leaves	Infusion	Orally	[67]
Malaria	Roots	Decoction	Orally	[67]
Epilepsy	Roots	Infusion and decoction	Orally	[68]
Respiratory problems	Roots	Powder Roots are dried and pulverized into powder and mixed into boiling water.	Inhalation	[68]
Joint pain	Roots	Pounded into paste	Poultice	[68]
Ringworm and fungal infections	Bark	Powder or Pounded paste	Topical application	[68]
Typhoid fever	Roots	Decoction	Orally	[68]
Rheumatism	Bark	Powder	Topical application	[68]
Asthma	Stem and bark	Tincture	Inhalation	[69]
Complementary medication for Cancer	Leaves	Infusion	Orally	[69]
Kidney infections and Diuretics	Leaves	Infusion and decoction	Orally	[70]

## Data Availability

All the used data are contained in this article.

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
