# Peer review of "Review of Mimusops zeyheri Sond. (Milkwood): Distribution, Utilisation, Ecology and Population Genetics"

_plants, 2024, doi:10.3390/plants13202943_

Round 1
Reviewer 1 Report
Comments and Suggestions for Authors
This is quite well written but interesting review paper in which the actual knowledge on the Mimusops zeyheri Sond. is presented.
In this manuscript the distribution, taxonomy, phytochemistry, ethnomedical applications, ecological functions, genetics diversity and biotechnological potential is discussed.
However, the manuscript is not very well prepared and requires significant improvement. Therefore I have some comments and suggestions.
My comments are as follows:
- The division into materials and methods and results in the review work is something strange. Therefore the manuscript should be reworded.
- The information that commonly used programs were used to prepare and select materials is strange, to say the least
- The list of cited works is prepared, to put it mildly, carelessly and requires significant improvement.
And some small comments and corrections:
- It is not allowed to specify: “…reported by [38]. [42} reported…”, or “…study by [51]. That means that number is reported or studying. There are many such strange expressions in the text and they need to be changed
- On page 12 line 534 is information: “…while Omotoyo et al. (2020)”? A citation number is needed.
- Magazine names are written sometimes with a capital letter and other times with a lowercase letter, except for the first word.
- There is no publisher in the cited textbooks and monographs
In conclusion, the manuscript needs to be substantially improved, before assessment about its suitability for publication in Plants is made.
Author Response
Comment: This is quite well well-written but interesting review paper in which the actual knowledge on the Mimusops zeyheri Sond. is presented.
Response: Thank you for recognizing the quality of writing and the interesting nature of our review paper. We're pleased that you found our presentation of current knowledge on Mimusops zeyheri Sond. to be informative and engaging.
Comment: In this manuscript, the distribution, taxonomy, phytochemistry, ethnomedical applications, ecological functions, genetics diversity and biotechnological potential are discussed.
Response: We appreciate your thorough overview of the topics covered in our manuscript. We aimed to provide a comprehensive review of Mimusops zeyheri Sond., and we are glad that you have noted the breadth of our coverage, from distribution and taxonomy to its biotechnological potential.
Comment: However, the manuscript is not very well prepared and requires significant improvement. Therefore I have some comments and suggestions.
Response: We value your constructive feedback and appreciate the time you have taken to provide comments and suggestions. We welcome the opportunity to improve our manuscript and look forward to addressing your specific points to enhance the quality and clarity of our work.
Comment: The division into materials and methods and results in the review work is something strange. Therefore the manuscript should be reworded.
Response: Thank you for the comment, the manuscript was reworded and the results were removed leaving discussed relevant sections.
Comment: The information that commonly used programs were used to prepare and select materials is strange, to say the least
Response: We acknowledge and appreciate the comment, but it is not clear what suggestion is being made. We, however, reviewed the content where the comment was made and could not make any changes.
Comment: The list of cited works is prepared, to put it mildly, carelessly and requires significant improvement.
Response: Thank you for the comment, the cited work has been reviewed extensively and corrections have been made accordingly.
Comment: It is not allowed to specify: “…reported by [38]. [42} reported…”, or “…study by [51]. That means that the number is reported or studying. There are many such strange expressions in the text and they need to be changed
Response: Thank you for the comment, The expressions, and sentences with references as numbers have been revisited and reworded for proper wording and referencing.
Comment: On page 12 line 534 is information: “…while Omotoyo et al. (2020)”? A citation number is needed
Response: Thank you for the comment. The citation number has been added and the citation has therefore been fixed.
Comment: Magazine names are written sometimes with a capital letter and other times with a lowercase letter, except for the first word.
Response: Thank you for the comment, The inconsistency in writing with lower and uppercase for megazines has been fixed.
Comment: There is no publisher in the cited textbooks and monographs
Response: Thank you for the comment, the publishers of the textbooks and monographs have been inserted. See references 6,7, 9 and 12 (Monograms), 1,2,3,4,5,8,10,11,13, 14,15, 16, 43, 44,54, 67, 74,82, 57 (Books)

Reviewer 2 Report
Comments and Suggestions for Authors
Dear Authors
thank you for providing me the Review article draft entttled "Review of Mimusops zeyheri Sond. (Milkwood): Distribution, Phytochemistry, Utilisation, Ecology and Population genetics" for my consideration
After having searched the provided data together with the outcome of my personal search I can confirm you that is a well organized Review article, which to my opinion has selected a not adequately studied endemic african plant
I can see that out of the "repeated" articled from several among the authors recent publications, there is a lack on Phytochemical studies, so out of the ethno and tradiditonal studies cannot support an intense Review article
It is noteworthy that the recent references (22- Perceptions on utilization, population, and factors that affecting local distribution of Mimusops zeyheri in the Vhembe Biosphere Reserve, S Africa. 2023 - by Lubisi et al. in reality has among authors Lubisi, Ramarumo, Manyaga, Mbeng and Mokghele and ref 51 (by Ethnobotanical Investigation of Mimusops zeyheri, an Underutilized Indigenous Fruit Tree in Gauteng Province, S Africa. Sustainability, 2024 by Matlala et al, has among authors Lubisi, Ndhlovu, Mokgehle , Otang-Mbeng
So, it is clear that several among you as authors have already two very recent publications (2023, 2024) on Mimusops zeyheri with more or less comparable data and similar conclusions of the need for further phytochemical studied on a species which has never been studied adequately.
Without these missing studies is of low interest to go on with further Reviews while there is no knowledge od the chemical profile of several different parts 9at least the most used, or most promissing for further exploitations)
There are more than 100 references on Mimusops zeyhleri in the text, where all have to be used with italics 9Lines, 159, 161, 164, 190, 194, 196, 198, 256, 259, 260, 262, 263, 266, 270, 273, 275, 277, 278 etc)
I would propose to the team of authors to complete the phytochemical studies on the plant species rather than submitting further reviews drafts, especially if the intend to expect and plan future exploitation of it.
The existing data on M. zeyhleri have been publisehd while another one has to be added (by Rmavhale et al 2024 DOI
10.4102/jomped.v8i1.238
while, the missing phytochemical studies have to be plannes, scheduled and presented so far, if a full Review is really intended to be planned
Unfortunatelly at this stage the OverReview is uncomplete to be named Review for this species without chemical data and based only to some nutritional values
Expect to further experimental phytochemical work on this very important species
Kind regards
Author Response
Dear Reviewer 2,
We extended our gratitude and expect to improve the manuscript. All the concerns that were made, we have extensively articulated on them and revised the manuscript.
R2
Comment: After having searched the provided data together with the outcome of my search I can confirm that this is well-organized. Review article, which to my opinion has selected a not adequately studied endemic African plant.
Response: We are truly grateful for your thorough evaluation of our manuscript, including your research efforts. Your confirmation that this is a well-organized review article is highly encouraging. We are particularly pleased that you recognize the significance of our choice to focus on Mimusops zeyheri Sond., an endemic African plant that has not received adequate scientific attention. Your comment validates our aim to contribute meaningful research to the field by shedding light on understudied yet potentially valuable plant species. We appreciate your recognition of both the quality of our work and the importance of the subject matter we've chosen to explore.
Comment: It is noteworthy that the recent references (22- Perceptions on utilization, population, and factors that affect the local distribution of Mimusops zeyheri in the Vhembe Biosphere Reserve, S Africa. 2023 - by Lubisi et al. in reality has among authors Lubisi, Ramarumo, Manyaga, Mbeng and Mokghele and ref 51 (by Ethnobotanical Investigation of Mimusops zeyheri, an Underutilized Indigenous Fruit Tree in Gauteng Province, S Africa. Sustainability, 2024 by Matlala et al, has among authors Lubisi, Ndhlovu, Mokgehle, Otang-Mbeng.
So, it is clear that several among you as authors have already two very recent publications (2023, 2024) on Mimusops zeyheri with more or less comparable data and similar conclusions of the need for further phytochemical studies on a species which has never been studied adequately.
Without these missing studies is of low interest to go on with further Reviews while there is no knowledge of the chemical profile of several different parts at least the most used, or most promising for further exploitations)
Response: Thank you for your thoughtful comments on our manuscript. We appreciate your attention to detail and would like to address the points you've raised:
- Firstly, we'd like to clarify that the papers you mentioned (Lubisi et al., 2023 and Matlala et al., 2024) are not review papers, but original research articles. These studies were conducted in two different provinces of South Africa, each focusing on specific aspects of Mimusops zeyheri in distinct geographical areas. The current manuscript is, in fact, our first review paper on this plant.
- The necessity for this review paper stems from several factors:
- a) Synthesis of Existing Knowledge: While our previous studies provided valuable data from specific regions, this review aims to synthesize information from all available sources on M. zeyheri, offering a comprehensive overview of the current state of knowledge.
- b) Identification of Research Gaps: By consolidating existing research, including our recent studies, this review helps identify critical gaps in our understanding of zeyheri, particularly in phytochemistry, which can guide future research efforts.
- c) Broader Contextual Understanding: This review places the findings from individual studies, including ours, within a wider context, allowing for a more holistic understanding of zeyheri's potential and challenges across different regions.
- d) Interdisciplinary Perspective: By bringing together ecological, ethnobotanical, and preliminary phytochemical data, this review aims to provide a multifaceted perspective that individual research papers cannot offer.
- e) Foundation for Future Studies: We believe this review will serve as a valuable resource for researchers, potentially catalyzing the much-needed phytochemical studies that we all agree are necessary.
- Regarding the phytochemical studies, we wholeheartedly agree with your assessment of their importance. Our review explicitly highlights this critical research gap and aims to stimulate interest in conducting these essential studies.
Comment: There are more than 100 references on Mimusops zeyheri, in the text, where all have to be used with italics Lines, 159, 161, 164, 190, 194, 196, 198, 256, 259, 260, 262, 263, 266, 270, 273, 275, 277, 278 etc)
Response: Thank you for the comment, the words Mimusops zeyheri and M. zeyheri as referenced in the article have been corrected and written in italics across the entire document.
Comment: I would propose to the team of authors to complete the phytochemical studies on the plant species rather than submitting further review drafts, especially if the intend to expect and plan future exploitation of it.
Response: We sincerely appreciate your suggestion regarding the importance of conducting phytochemical studies on M.zeyheri. We are in full agreement with your perspective on the critical nature of such research for advancing our understanding of this species, particularly about its potential future exploitation.
We are pleased to inform you that our research team is currently engaged in a comprehensive phytochemical study of M. zeyheri. This ongoing research includes:
- Detailed chemical profiling of various parts of the plant, with a focus on those most commonly used in traditional practices and those showing promise for potential applications.
- Investigation of bioactive compounds and their properties.
This study is well underway, and we anticipate publishing the results in a separate, dedicated paper. We aim to address the very gap in knowledge that you have rightly identified as crucial for the scientific community.
Comment: The existing data on M. zeyheri, have been published while another one has to be added (by Rmavhale et al 2024 DOI 10.4102/jomped.v8i1.238
Response: Thank you for recommending the addition of the study in the review. The study has been reviewed and added to the article. The study by Ramavhale appears in Table 3.1 and the document as reference 56.

Reviewer 3 Report
Comments and Suggestions for Authors
This is a fairly interesting paper, in which the authors document that the review on Mimusops zeyheri Sond. (Milkwood), an indigenous fruit tree species in with considerable ecological, cultural, and nutritional. The review shows current knowledge of its distribution, taxonomy, phytochemistry, ethnomedicinal applications, ecological functions, genetic diversity, and biotechnological potential. It has a clear and straightforward design. Thus, it might be sufficient for publication up to the ''plants'' criteria. Finally, I decide to accept this manuscript, but the authors must better explain the phytochemical part if there are studies on the isolation of chemical compounds and their biological activity
Author Response
R3
Comment: This is a fairly interesting paper, in which the authors document that the review on Mimusops zeyheri Sond. (Milkwood), an indigenous fruit tree species in with considerable ecological, cultural, and nutritional. The review shows current knowledge of its distribution, taxonomy, phytochemistry, ethnomedicinal applications, ecological functions, genetic diversity, and biotechnological potential. It has a clear and straightforward design. Thus, it might be sufficient for publication up to the ''plants'' criteria. Finally, I decide to accept this manuscript, but the authors must better explain the phytochemical part if there are studies on the isolation of chemical compounds and their biological activity
Response: Thank you for your feedback and acceptance of the manuscript. We appreciate your recognition of the comprehensive nature of the review on Mimusops zeyheri Sond. and its clear and straightforward design. Regarding the phytochemical aspects, we acknowledge your recommendation to better explain this part of the review. There are currently no reported studies on the phytochemical analysis of the Reviewed plant. This gap has been identified and discussed in the paper. The authors are working on a separate paper aimed at conducting comprehensive phytochemical analysis of the plant species.

Round 2
Reviewer 1 Report
Comments and Suggestions for Authors
The manuscript was revised taking into account most of my corrections, suggestions and comments and other reviewers comments.
Based on the above, I can recommend this version of paper for publication in presented form.
Author Response
Thank you for your feedback and acceptance of the manuscript. We appreciate your recognition of the comprehensive nature of the review on Mimusops zeyheri Sond.
Reviewer 2 Report
Comments and Suggestions for Authors
Dear Authors
thank you for providing me the revised version of your Review article
My opinion remains unchanged to wards this submission. I cannot find a robust reason for a Review on a plant , proposing potential exploitation, based only on phytochemical basics. such as TPC and antioxidative activity.
The fingerprint to the scientific Community is really very lowt, that is the reason I insist on rejecting it, and provoke you to step forward to real phytochemical studies.
Without knowledge of the chemical profile, a review on the species is really of low interest to be based only to Diversity as well as on historical and traditional uses, while is well known that many traditionally used plants have proved inconvenient for potential human use, due to their content in particular metabolites
Unfortunately I cannot send you a positive outcome
Kind regards
Author Response
Thank you for the positive contribution related the this manuscript. We sincerely appreciate your suggestion regarding the importance of conducting phytochemical studies on M.zeyheri. We are in full agreement with your perspective on the critical nature of such research for advancing our understanding of this species, particularly about its potential future exploitation. We are pleased to inform you that our research team is currently engaged in a comprehensive phytochemical study of M. zeyheri. This ongoing research includes:
- Detailed chemical profiling of various parts of the plant, with a focus on those most commonly used in traditional practices and those showing promise for potential applications.
- Investigation of bioactive compounds and their properties.
This study is well underway, and we anticipate publishing the results in a separate, dedicated paper. We aim to address the very gap in knowledge that you have rightly identified as crucial for the scientific community.
Synthesis of Existing Knowledge: While our previous studies provided valuable data from specific regions, this review aims to synthesize information from all available sources on M. zeyheri, offering a comprehensive overview of the current state of knowledge.
Identification of Research Gaps: By consolidating existing research, including our recent studies, this review helps identify critical gaps in our understanding of zeyheri, particularly in phytochemistry, which can guide future research efforts.
Broader Contextual Understanding: This review places the findings from individual studies, including ours, within a wider context, allowing for a more holistic understanding of zeyheri's potential and challenges across different regions.
Interdisciplinary Perspective: By bringing together ecological, ethnobotanical, and preliminary phytochemical data, this review aims to provide a multifaceted perspective that individual research papers cannot offer. Foundation for Future Studies: We believe this review will serve as a valuable resource for researchers, potentially catalyzing the much-needed phytochemical studies that we all agree are necessary.